# Association between the Strength of Flexor Hallucis Brevis and Abductor Hallucis and Foot Mobility in Recreational Runners

Antonio C. F. Andrade [1], Danilo S. Catelli [2,*], Bruno L. S. Bedo [3], Guilherme M. Cesar [4], Thiago F. Santos [5], Eduardo B. Junqueira [5] and Paulo R. P. Santiago [5,*]

1. Program in Rehabilitation and Functional Performance, Ribeirão Preto Medical School, University of São Paulo, Ribeirão Preto 14049-900, SP, Brazil
2. Human Movement Biomechanics Research Group, Department of Movement Sciences, KU Leuven, 3001 Leuven, Belgium
3. Department of Sport, School of Physical Education and Sport, University of São Paulo, São Paulo 05508-030, SP, Brazil
4. Department of Physical Therapy, University of North Florida, Jacksonville, FL 32224 , USA
5. School of Physical Education and Sport of Ribeirão Preto, University of São Paulo, Ribeirão Preto 14040-907, SP, Brazil
* Correspondence: dan.catelli@kuleuven.be (D.S.C.); paulosantiago@usp.br (P.R.P.S.)

**Abstract:** Different measurements of foot morphological characteristics can effectively predict foot muscle strength. However, it is still uncertain if structural and postural alterations leading to foot pronation could be compensated with more efficient function of the intrinsic foot muscles and how mobility and strength are associated. Additionally, the relationship between foot mobility and the strength of the intrinsic muscles that control the foot arch is still unclear. Therefore, this study aimed to investigate the morphological parameters between dominant and non-dominant feet and the relationship between the intrinsic foot muscle strength and foot mobility in recreational runners. We used a cross-sectional study design to evaluate twenty-four healthy recreational runners (minimum 15 km/week) with an average training history of $70 \pm 60$ months. Foot Posture Index (FPI-6), isometric intrinsic muscle strength, overall morphology, and normalized mobility of both feet were assessed. Parametric tests analyzed the unidimensional measures, and paired analysis determined differences between dominant and non-dominant sides. Pearson's and Spearman's correlation coefficients determined the relationships between normalized strength and the variables of interest (CI = 95%). There was no significant association between intrinsic foot muscle strength and mobility. The only difference observed was between the dominant and non-dominant foot regarding the normalized foot length and midfoot width during non-weight-bearing, with small and medium effect sizes, respectively. Neither foot morphology nor foot mobility was associated with strength from intrinsic foot muscles in healthy recreational runners. Further work should explore the relationship investigated in our study with professional athletes and runners with symptomatic lower limb injuries to potentialize training and rehabilitation protocols.

**Keywords:** strength; isometric force; running; foot forces; intrinsic musculature

## 1. Introduction

Running-related musculoskeletal pain affects 22% of runners [1]. The ankle–foot complex, in particular, represents 26% of all these injuries reported by runners [2]. As the first component of the kinetic chain to touch the ground during running, the human foot has to be stiff enough during foot-strike and push-off, and mobile/compliant enough during the stance phase [3]. In addition, the leg swing is also fundamental to running performance. The swinging limbs can improve performance in jumping and running, since it is conceivable that runners could use their leg swing to increase vertical ground force in the same way as arm swing in vertical jumping [4].

The human foot is a very complex structure. The arrangement between bones, ligaments and joint capsules creates the arch of the foot [3], and these structures configure the morphology of the foot as a half dome. Still, simple measurements of foot morphological characteristics can effectively predict foot muscle strength [5]. Traditionally, the longitudinal medial and transverse arches are the most studied. A recent investigation of the transverse arch in vitro demonstrated that intermetatarsal tissues and metatarsal mobility influenced medial longitudinal stiffness of the foot [6]. The medial longitudinal arch also plays an important role in locomotion, and it supports the body weight in standing posture and dynamic movements [7]. Furthermore, the medial longitudinal arch can allow for energy efficient running by acting as a spring (or a rubber ball) absorbing and storing elastic strain energy from the body during the first half of the stance phase of running and returning it during the second half of stance [8]. Therefore, pronation is a necessary function in gait. However, if done excessively, the structural and morphological alterations in the longitudinal arch can lead to greater pronation of the foot, and therefore, dysfunctions. As an example, flat foot can lead to poor motor skills and physical performance in children [9]. Individuals with patellofemoral pain syndrome have more foot mobility compared with matched controls [10]. This augmented mobility leads to a more pronated foot, causing kinematic alterations in the lower limb kinetic chain [11].

Although the theory supporting pain due to foot postural alterations predominated for a long time, it is known that some individuals with pronated foot are asymptomatic and lower-limb-injury-free. Recent studies have demonstrated that people with pronated feet and lower limb injuries have a smaller cross-sectional area of intrinsic foot muscles compared with those with pronated feet but who are injury-free in the lower limbs [12]. Furthermore, some studies also have demonstrated that individuals with patellofemoral pain syndrome with greater midfoot mobility treated with foot orthoses and hip exercises demonstrate similar outcomes [13]. Based on these observations, is still uncertain if structural and postural alterations leading to foot pronation could be compensated with more efficient function of the intrinsic foot muscle strength, and how mobility and strength are associated in the foot.

The intrinsic muscles of the feet are active in dorsal and plantar functions. The intrinsic plantar muscles are commonly described based on their functional links with the longitudinal and transverse arches [14]. These muscles are composed of four layers of deep muscles in the plantar aponeurosis. The first two layers have muscle configurations that align with the foot's medial and lateral longitudinal arches [3]. Therefore, the intrinsic plantar muscles are functionally linked with the transverse and longitudinal medial arches [15]. Although the intrinsic muscles are more active in dynamic activities (e.g., walking) than standing [16], training these muscles enhances foot posture [17]. In addition, intrinsic foot muscles provide dynamic arch support during the propulsive phase of gait [18] and support the foot as a platform for standing and a lever for propelling the body during dynamic activities [19]. When considering running, the intrinsic muscles can stiffen the longitudinal arch during the push-off phase [20] to control foot posture [21]. Further, these muscles can guarantee an efficient transference of power generated by the ankle flexors soleus and gastrocnemius.

Weakness of the plantar intrinsic foot muscle is observed in individuals with plantar fasciopathy [22]. Physical activity levels and footwear use may influence intrinsic foot muscles and mobility. Kenyan adolescents accustomed to barefoot running and engaged in more vigorous physical activity have stronger intrinsic foot muscles and greater foot mobility than matched controls accustomed to shod running. The prevalence of injuries in the habitually barefoot was 8% versus 61% among the accustomed shod [23]. Strengthening exercise programs for intrinsic foot muscles can improve performance in physical tests [24,25]. From these observations, there seems to be a correlation between intrinsic foot muscle strength, foot mobility and running performance.

The questions above show the possibility of an association between foot mobility and the strength of the intrinsic muscles that control the foot's arch. In clinical practice,

professionals deal with situations requiring approaches to increase mobility, increase intrinsic foot muscle strength or use foot orthoses. This dichotomous relationship may cause difficulties for sports coaches, physical therapists, runners and people involved in running activities to achieve the best strategies to prevent and treat lower-limb injuries. In recreational runners free from injuries, this relationship is not fully understood. This cross-sectional study aimed to investigate whether an association exists between intrinsic foot muscle strength and foot mobility in recreational runners who are free from injuries.

## 2. Materials and Methods

### 2.1. Participants

This cross-sectional and correlational study investigated 24 (18 men and 6 women) recreational runners selected via social media announcements. In addition, the principal investigator performed previous telephone screenings of potentially eligible participants. Study inclusion criteria involved 18–45 years of age; running at least 15 km/week; and no history of ligament laxity, meniscal pathology, patellar tendonitis, knee pain from acute trauma, patellar dislocation, or lower-limb surgeries. Women should not be pregnant during the data acquisition. The Ethics Committee approved the study, which was performed according to the Declaration of Helsinki. All participants provided written informed consent to participate.

### 2.2. Procedures

Twenty-four participants were included in the study (Table 1). The dominant side was defined as the preferred side for kicking a soccer ball. Both legs were assessed in random order [26] while the participants had bare feet and were wearing shorts. Given partial data collection occurred during the COVID-19 pandemic, all biosafety criteria were strictly followed.

**Table 1.** Summary of participants' demographics, reporting mean (standard deviation).

| Parameters | Participants (n = 24) |
|---|---|
| Age (years) | 33.58 (7.13) |
| Sex (female n (%)) | 6 (25%) |
| Height (m) | 1.71 (0.07) |
| Weight (kg) | 70.30 (10.28) |
| BMI (kg/m$^2$) | 23.84 (2.54) |
| Training Experience (months) | 70 (60) |
| Leg dominance (right n (%)) | 19 (79.17) |

### 2.3. Static Foot Posture

Several methods have been reported to quantify or classify standing foot posture [27], and the Foot Posture Index (FPI) has been proposed as a fast, simple method of visually classifying foot posture as either pronated, supinated or expected based on six different visual foot posture criteria [28]. The FPI-6 is composed of a 6-item assessment and classification tool for foot posture. It is performed with the participant in a relaxed stance with weight distributed equally on both feet. Scores range from −2 to +2; the total score is between −12 (highly supinated) and +12 (highly pronated) [29]. Thus, we classified participants' feet as supinated (−12 to −1), normal (0 to +5), or pronated (+6 to +12). The FPI is intended to be a simple method of scoring the various features of foot posture into a single quantifiable result, which in turn gives an indication of the overall foot postures.

### 2.4. Flexor Hallux Strength

The force of the muscles that control the medial longitudinal arch of the foot was measured with a load cell linked with an Arduino (an open-source electronic prototyping platform) for the signal acquisition of force in kilograms. We used a custom-made foot strength measurement, a wooden platform with a setup to stabilize the ankle–foot complex,

as seen in Figure 1, according to the device used by Quek et al. (2015) [30], who verified excellent intra-rater reliability (ICC = 0.982, CI = 0.96–0.99).

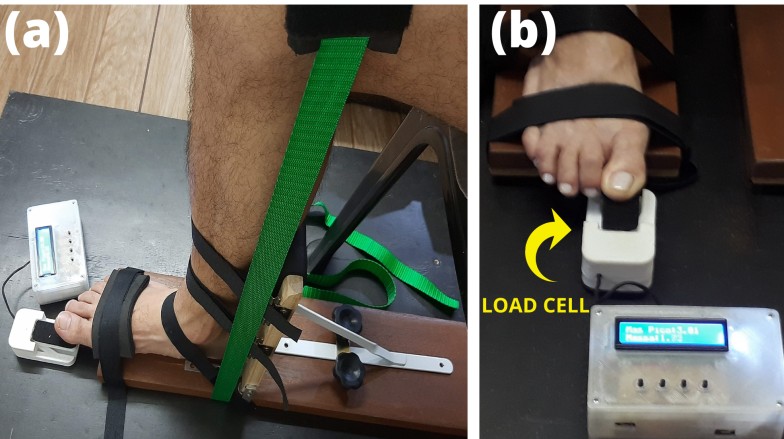

**Figure 1.** Measurement of hallux flexor muscles with a wooden platform: (**a**) setup to stabilize the ankle–foot complex and knee; (**b**) load cell.

Participants sat on a chair with a backrest and maintained an angle of 90° between ankle–foot/shank, shank/thigh, and thigh/trunk. Arms remained crossed in front of the chest. The forefoot, ankle, and distal shank were stabilized with Velcro straps to avoid joint movements. Another inelastic strap was used to stabilize the knee. A load cell was aligned to the plantar surface of the hallux. Participants were instructed to press the hallux towards the ground, lifting the foot's plantar arch. Participants were not allowed to flex the interphalangeal joint of the hallux, adduct the hallux, raise the heel, engage the hamstrings, nor push the chair back. They trained with this task until they were familiarized and correctly performed the movement. A maximal isometric force was collected for 5 s three times, with a 30 s rest between trials. The maximum hallux flexor force capacity was normalized by body weight times body height, compounding the normalized foot strength [31].

*2.5. Foot Structure and Mobility*

We measured foot structure and mobility with a method previously described and validated [32] (Figure 2). The participant stood on a custom-made wooden platform with the heels 15 cm apart and weight-bearing equally distributed on both feet. The distance of the most posterior aspect of the calcaneus to the most distal point of the foot (first or the second toe, depending on which was longer) represented the foot length. This foot length was normalized by the participant's height [33,34]. Therefore, the foot dorsum was demarcated as 50% of the foot length. At this point of the midfoot, we measured the foot arch height (FAH) and midfoot width (MFW) in terms of weight bearing (WB) with a combination square and with a modified caliper, respectively. The participant was then seated on an examination plinth with the legs hanging freely off the edge. A plastic portable platform with 80-grit sandpaper touched the plantar surface of the foot and was used to support the combined square for the measurement of the foot arch height. The same measures made during weight-bearing condition were retaken during non-weight-bearing (NWB). Foot mobility was determined by: (i) foot arch height in WB subtracted from foot arch height in NWB (DiffFAH); (ii) midfoot width in NWB subtracted from midfoot width in WB (DiffMFW). General foot mobility was represented by the composition of the vertical and medio-lateral mobility and was obtained as:

$$\sqrt{((DiffFAH)^2 + (DiffMFW)^2)} \qquad (1)$$

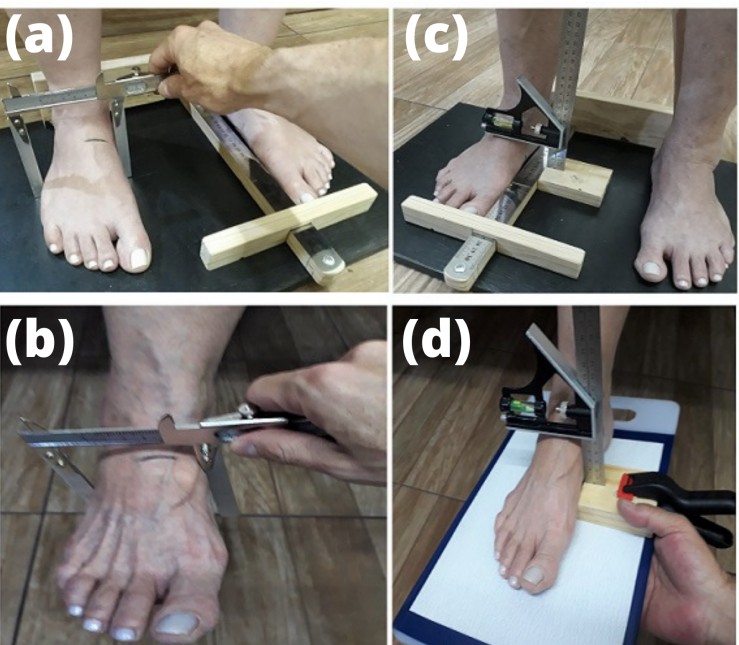

**Figure 2.** Morphology and anthropometric measurements to quantify the foot mobility magnitude [32]: (**a**) midfoot width, weight-bearing; (**b**) midfoot width, non-weight-bearing; (**c**) foot arch height, weight-bearing; (**d**) foot arch height, non-weight-bearing.

*2.6. Statistical Analysis*

Descriptive statistics were performed. The Shapiro–Wilk test determined the normality of the data. Histograms and boxplots were created for exploratory data analysis and identification of outliers. The Foot Posture Index (FPI-6) is a tool with ordinal items that cannot be analyzed with statistical parametric tests. We used Rasch analysis to transform ordinal scores obtained by adding scores from each item of FPI-6 into interval measures with "logit values" [35]. Parametric tests were used to analyze these unidimensional measures. For independent analysis, data from dominant and non-dominant limbs were included in different data sets. Paired sample T-tests determined differences between dominant and non-dominant sides. Pearson's and Spearman's correlation coefficients determined the relationships between normalized strength and variables such as general foot mobility, normalized foot length, foot height, and foot width (in weight-bearing posture); and the difference in foot arch height and midfoot width in weight- and non-weight-bearing situations ($p < 0.05$). All analyses were performed using JASP software (version 0.14).

**3. Results**

Demographics and characteristics of the participants are presented in Table 1. The distribution of foot posture according to FPI-6 for both sides were: supinated (n = 2 dominant and n = 3 non-dominant), normal (n = 20 dominant and n = 19 non-dominant), and pronated (n = 2 dominant and n = 2 non-dominant).

There were no significant differences between dominant and non-dominant sides (Table 2) for Foot Posture Index (95% CI: −0.098 to 0.418; $p = 0.213$), normalized foot strength (95% CI: −0.142 to 0.673; $p = 0.201$), foot arch height (WB) (95% CI: −0.091 to 0.731, $p = 0.127$), midfoot width (WB) (95% CI: −0.129 to 0.687, $p = 0.180$), foot arch height (NWB) (95% CI: −0.218 to 0.589, $p = 0.368$), DiffFAH (95% CI: −0.519 to 0.284, $p = 0.567$), DiffMFW (95% CI: −0.542 to 0.262, $p = 0.494$), or general foot mobility (95% CI: −0.757 to 0.068, $p = 0.101$). There was a statistical difference for normalized foot length (95% CI: −0.129 to −0.008, $p = 0.029$, effect size = −0.475) and midfoot width (NWB) (95% CI: 0.029 to 0.212, $p = 0.012$, effect size = 0.558)—Figure 3.

**Table 2.** Characteristics of the foot posture and mobility for the dominant and non-dominant sides.

| Parameters | Dominant Mean (SD) | Non-Dominant Mean (SD) | *p*-Value | 95% CI [Lower, Upper] |
|---|---|---|---|---|
| Foot Posture Index (FPI-6) | 1.16 (1.45) | 1.00 (1.42) | 0.213 | [−0.098, 0.418] |
| Normalized Foot Length | 14.99 (0.47) | 15.06 (0.44) | 0.029 | [−0.129, −0.008] |
| Normalized Foot Strength | 12.63 (3.73) | 12.38 (3.90) | 0.201 | [−0.147, 0.658] |
| Foot Arch Height (cm)—WB | 6.72 (0.47) | 6.64 (0.55) | 0.127 | [−0.024, 0.183] |
| Midfoot Width (cm)—WB | 8.57 (0.62) | 8.50 (0.61) | 0.180 | [−0.035, 0.1771] |
| Foot Arch Height (cm)—NWB | 7.82 (0.42) | 7.77 (0.54) | 0.368 | [−0.063, 0.163] |
| Midfoot Width (cm)—NWB | 7.28 (0.52) | 7.16 (0.53) | 0.012 | [0.029, 0.212] |
| DiffFAH (cm) | 1.10 (0.23) | 1.13 (0.25) | 0.567 | [−0.133, 0.075] |
| DiffMFW (cm) | 1.29 (0.37) | 1.34 (0.34) | 0.494 | [−0.199, 0.099] |
| General Foot Mobility (cm) | 1.73 (0.29) | 1.79 (0.26) | 0.101 | [−0.135, 0.013] |

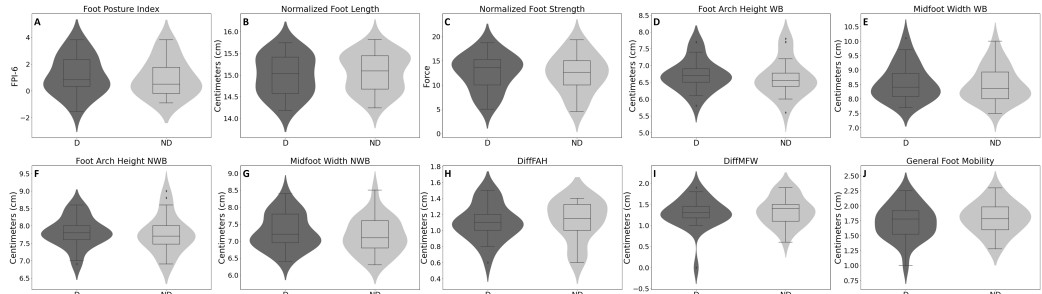

**Figure 3.** Boxplots and histograms of the variables: (**A**) Foot Posture Index (FPI-6); (**B**) normalized foot length; (**C**) normalized foot strength; (**D**) foot arch height—WB; (**E**) midfoot width—WB; (**F**) foot arch height—NWB; (**G**) midfoot width—NWB; (**H**) DiffFAH; (**I**) DiffMFW; and (**J**) general foot mobility. The violin plots show the data distribution, and the boxplots present the data locality, spread and skewness of dominant (black) and non-dominant (gray) feet.

Foot height and width for the dominant side and foot height for the non-dominant side revealed outliers in a previous exploratory analysis. The correlations between normalized strength and other variables are presented in Table 3. There was no significant correlation between normalized strength and the general foot mobility, normalized foot length, the foot height and width (in weight-bearing), or the difference between foot arch height and midfoot width between weight-bearing and non-weight-bearing settings.

**Table 3.** Correlations between general foot mobility, normalized foot length, foot height, foot width, DiffFAH, DiffMFW, and normalized foot strength for dominant and non-dominant sides.

| Parameters | Normalized Strength (Dominant) Value | *p*-Value 95% | Normalized Strength (Non-Dominant) Value | *p*-Value 95% |
|---|---|---|---|---|
| General Foot Mobility (cm) | 0.063 (r) | 0.771 [−0.349, 0.455] | 0.119 (r) | 0.579 [−0.299, 0.499] |
| Normalized Foot Length | −0.076 (r) | 0.723 [−0.338, 0.465] | 0.117 (r) | 0.587 [−0.301, 0.497] |
| Foot Arch Height (WB) (cm) | −0.224 (rho) | 0.292 [−0.575, 0.197] | −0.095 (rho) | 0.658 [−0.480, 0.320] |
| Midfoot Width (WB) (cm) | 0.134 (rho) | 0.532 [−0.285, 0.510] | 0.146 (rho) | 0.495 [−0.273, 0.519] |
| DiffFAH (cm) | 0.158 (rho) | 0.461 [−0.262, 0.528] | 0.145 (rho) | 0.500 [−0.275, 0.518] |
| DiffMFW (cm) | 0.124 (rho) | 0.565 [−0.294, 0.502] | −0.001 (rho) | 0.995 [−0.404, 0.402] |

Notes: *CI* confidence interval; *r* Pearson's correlation coefficient; *rho* Spearman's correlation coefficient; DiffFAH (difference in foot arch height); DiffMFW (difference in midfoot width).

## 4. Discussion

This study aimed to investigate the differences between the dominant and non-dominant foot in the morphological parameters and the relationship between the intrinsic

foot muscles' strength and foot mobility in recreational runners. The main findings were (i) a difference between dominant and non-dominant feet in normalized foot length and midfoot width in the non-weight-bearing setting and (ii) correlations between normalized strength and the variables of interest.

An objective measure of the foot's intrinsic muscle strength is necessary to establish parameters for improvement in rehabilitation and performance programs. The measurement methods can be classified into: (i) indirect methods (which do not directly measure strength but provide information about the structure and activity of these muscles) and (ii) direct methods that measure the units of force [15]. Quek et al. (2015) measured hallux flexor strength with a Nintendo Wii Balance Board in seated individuals, and the results demonstrated excellent intra-rater reliability (ICC = 0.982) [30]. We used a similar measurement but with a load cell. Furthermore, we decided to use a custom-made foot strength measurement platform with a particular setup to stabilize the ankle–foot complex. This set-up avoids raising the heel by interfering with soleus and hamstring activities through knee flexion. We used extra inelastic straps to stabilize the forefoot, midfoot, and distal shank. This new setup was like that used by Allen and Gross (2003) to evaluate toe flexor strength in individuals with plantar fasciitis [22]. This author encountered the intraclass correlation coefficient values for intra-rater reliability equal to 0.99. We believe this stabilization method is essential to avoid underestimating the importance of strength development capacity. The mean values of our non-normalized results of strength are like those encountered by Allen and Gross, although it was a population with different characteristics.

Although it is difficult to isolate the muscles that influence certain movements, our study focused on measuring the strength of the flexor hallucis brevis and the abductor hallucis. Using ultrasound to measure the cross-sectional area of the abductor hallucis and flexor hallucis brevis, Latey et al. (2018) encountered a weak positive correlation between the cross-sectional area of abductor halux and toe strength measured with hand-held dynamometry and a strong and significant correlation with pedobarography [36]. The correlation between the cross-sectional area of flexor hallucis brevis and strength of this muscle measured by hand-held dynamometry and pedobarography was weak and non-significant. This study did not investigate correlations of hallucis brevis size or strength with general foot mobility. Koyama et al. (2019) found a significantly greater intrinsic foot muscle strength in judo athletes than matched controls, but the thicknesses of all intrinsic foot muscles did not significantly differ between groups [37].

On the other hand, Xiao et al. (2020) encountered positive correlations between toe/metatarsophalangeal joint flexor strength and foot length, foot width, and truncated foot length [5]. This author used a different methodology than we used. The focus of this study was to correlate the foot's morphology and strength. The strength was measured with different methods. Our principal focus was on foot mobility, not only morphology.

While in our study the objective was to investigate the relationship between strength and mobility, our results are different from those of other studies whose objectives were to investigate the relationship between strength and morphology/structure. Despite these different objectives, the mobility findings of our study are similar to the results encountered by McPoil et al. (2009) [32].

The results of the present observational cross-sectional study raise the question of whether there is a relationship between strength and mobility. Some authors demonstrated that asymptomatic persons with lower limb injuries with pronated feet could have more thick intrinsic muscles compared to those with pronated feet and injuries in lower limbs [12]. In addition, the structure is not directly associated with the function, and mobility depends on tissue flexibility, joint movement (arthrokinematics), and motor control. As one of the causes of the pronated foot is elevated foot mobility, we decided to investigate this relationship with methods with easy use in practice clinics for recreational runners. This could compensate for structural deficits during tasks with weight bearing. Another important result of our study is emphasizing that based on experimental conditions, the structure/morphology is not associated with the performance of strength

development. Complementary work is being done in our laboratory to investigate the kinematic performance of these segments in this population. As an example of a training option, flexibility training, such as static stretching or proprioceptive neuromuscular facilitation stretching, should be adequate to enhance mobility [38].

Although our results in this population did not demonstrate an association between strength and mobility, further studies are necessary to elucidate this association in runners with musculoskeletal injuries in lower limbs, especially when considering muscle synergies in rehabilitation [39]. Patellofemoral pain syndrome is the most common injury in recreational runners. Higher foot mobility in this population leads to higher pronation. The in-shoe foot orthosis is the recommendation based on the consensus of specialists and is evidence-based for those cases. However, a recent study [13] demonstrated no significant difference between groups with greater foot mobility treated with exercises based on the hip approach and that match those treated with foot-in-shoe orthoses. It is necessary to investigate this relationship and evaluate if muscle strength is greater in this population as a compensatory mechanism. It is unlikely that intervention in one of these variables will be able to produce results in another in multimodal treatment plans to prevent and treat running injuries because of the weakness of the associations between these variables. Inferences should be made with caution.

Certain limitations can impact the results of our study. Although the protocol of force acquisition has been validated previously [30], we did not add loading of a half and/or the body weight to the participant. Therefore, future studies should include the participants' body weights to investigate the consequent deformation of their feet and observe foot mobility under more extensive loading. Furthermore, this study included men and women in the same sample size. Therefore, future investigations must also investigate the comparison between genders.

## 5. Conclusions

An association between intrinsic foot muscle strength and foot mobility in recreational runners free from lower extremity injuries was not observed in our study. However, our findings are limited to the characteristics of our cohort of runners. Further work with different populations of runners, such as professionals or symptomatic individuals, is necessary to validate our findings and to advance the knowledge to guide performance and care for runners.

**Author Contributions:** Conceptualization, A.C.F.A., G.M.C. and P.R.P.S.; methodology, A.C.F.A. and P.R.P.S.; software, A.C.F.A., T.F.S., B.L.S.B. and P.R.P.S.; validation, A.C.F.A., G.M.C. and P.R.P.S.; formal analysis, A.C.F.A., T.F.S., B.L.S.B., E.B.J. and P.R.P.S.; investigation, A.C.F.A., D.S.C., T.F.S., E.B.J., B.L.S.B., G.M.C. and P.R.P.S.; resources, A.C.F.A. and P.R.P.S.; data curation, A.C.F.A., D.S.C., T.F.S., E.B.J., B.L.S.B., G.M.C. and P.R.P.S.; writing—original draft preparation, A.C.F.A.; writing—review and editing, A.C.F.A., D.S.C., G.M.C. and P.R.P.S.; visualization, A.C.F.A., D.S.C., T.F.S., E.B.J., B.L.S.B., G.M.C. and P.R.P.S.; supervision, P.R.P.S.; project administration, P.R.P.S.; funding acquisition, A.C.F.A. and P.R.P.S. All authors have read and agreed to the published version of the manuscript.

**Funding:** This research was funded by Coordenação de Aperfeiçoamento de Pessoal de Nível Superior (CAPES-Brazil), grant number 001; Conselho Nacional de Desenvolvimento Científico e Tecnólogico (CNPq-Brazil), grant number 432259/2018-0; and Pro-Rectory of Research of University of São Paulo—USP ("Support Program for Projects that Make Use of Intelligent Digital Systems").

**Institutional Review Board Statement:** The study was conducted in accordance with the Declaration of Helsinki and approved by the Institutional Review Board of the School of Physical Education and Sport of Ribeirão Preto at the University of São Paulo (3.673.409, 2019).

**Informed Consent Statement:** Informed consent was obtained from all subjects involved in the study.

**Data Availability Statement:** Data can be made available upon request.

**Conflicts of Interest:** The authors declare no conflict of interest.

## Abbreviations

The following abbreviations are used in this manuscript:

| | |
|---|---|
| BMI | Body mass index |
| DiffFAH | Difference in foot arch height between weight- and non-weight-bearing conditions |
| DiffMFW | Difference in midfoot width between weight- and non-weight-bearing conditions |
| FAH | Foot arch height |
| FPI-6 | Foot Posture Index |
| ICC | Intraclass correlation coefficients |
| MFW | Midfoot width |
| NWB | Non-weight-bearing |
| WB | Weight-bearing |

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
