# Peer review of "Association between the Strength of Flexor Hallucis Brevis and Abductor Hallucis and Foot Mobility in Recreational Runners"

_2673-7078, doi:10.3390/biomechanics2040048_

Round 1

Reviewer 1 Report

This paper intends to investigate the important topic on the relationships between foot muscle strength and foot mobility in recreational runners who may sometimes suffer from foot injuries.  

1. Although the method itself employed to measure foot muscle strength in this study seems to be appropriate, the measurement condition is unlikely to sufficient to obtain some findings on the relationships between foot muscle strength and mobility. The authors should have added loading of a half and/or a body weight to the participants, which may have deformed their feet more and observed foot mobility under larger loading.

2. When calculating statistical parameters, the authors should have separated the male and female participants because there may be significant differences in foot strength and foot mobility between them. If so, the authors may have obtained different results and conclusions.

3. The authors should provide more detailed definitions, explanations and theoretical basis on foot mobility, foot posture index and general foot mobility which is original by the authors. 

Author Response

POINT

REVIEWER COMMENT

RESPONSE

Reviewer #1

1

Although the method itself employed to measure foot muscle strength in this study seems to be appropriate, the measurement condition is unlikely to sufficient to obtain some findings on the relationships between foot muscle strength and mobility. The authors should have added loading of a half and/or a body weight to the participants, which may have deformed their feet more and observed foot mobility under larger loading.

Thank you for the suggestion. We based the study's rationale on previous studies (e.g., Quek et al., 2015) where the participants were kept sit during the protocol. However, aiming to guide future investigations, we have inserted as a study limitation the force acquisition protocol and that future studies should quantify the foot strength while the feet are being loaded (half and/or a body weight).

Quek, J., Treleaven, J., Brauer, S. G., O'Leary, S., & Clark, R. A. (2015). Intra-rater reliability of hallux flexor strength measures using the Nintendo Wii Balance Board. Journal of Foot and Ankle Research, 8(1). https://doi.org/10.1186/s13047-015-0104-7

The limitation is now included in the text (page 8):

Certain limitations can impact the results of our study. Although the protocol of force acquisition has been validated previously (Quek, 2015), we have not added loading of a half and/or the body weight to the participant. Therefore, future studies should include the participant's body weight to investigate the consequent deformation of their feet and observe foot mobility under more extensive loading.

2

When calculating statistical parameters, the authors should have separated the male and female participants because there may be significant differences in foot strength and foot mobility between them. If so, the authors may have obtained different results and conclusions.

We appreciate this suggestion. The between-sex comparison has been previously described (Park et al., 2022), and we agree that future studies must also include this comparison. Unfortunately, given the sample size of our study (N=24), the appropriate statistical approach would not support a comparison between 18 men and 6 women. Since we also value the reviewer's comment and agree with between-sex comparisons for this type of research investigation, we included this issue in the limitation section.

3

The authors should provide more detailed definitions, explanations and theoretical basis on foot mobility, foot posture index and general foot mobility which is original by the authors.

We appreciate this critical comment. We reviewed the definitions of all variables described and edited the text (page 6). To clarify to the reviewer, foot mobility has been previously described in the literature, and several different methods have been related to quantifying or classifying standing foot posture (Cornwall & McPoil, 2011). The Foot Posture Index (FPI) has been proposed as a fast, simple method of visually classifying foot postures as either pronated, supinated or expected based on six different visual foot posture criteria (Redmond, 1998). The FPI used three variables; Difference in Dorsal Arch Height (DAHDIFF), Difference in Midfoot Width (MFWDIFF) and Foot Mobility Magnitude (FMM). The dorsal arch height in weight bearing is subtracted from the dorsal arch height measured in a non-weight direction to determine the DAHDIFF. The midfoot width measured in a non-weight approach was subtracted from the midfoot width obtained in weight bearing to determine the MFWDIFF. The FMM is a composite measure of both DAHDIFF and MFWDIFF and involves taking the square root of the sum of each variable after it has been squared (Cornwall & McPoil, 2011; Redmond, 1998).

Cornwall, M.W., & McPoil, T.G. (2011). Relationship between static foot posture and foot mobility. Journal of Foot and Ankle Research, 4(1). (doi: 10.1186/1757-1146-4-4).

Redmond, A. (1998). THE FOOT POSTURE INDEX © Easy quantification of standing foot posture Six item version FPI-6 USER GUIDE AND MANUAL AUGUST 2005 Foot Posture Index-User guide and manual 2 Acknowledgments The FPI was developed with funding from the following agencies. www.leeds.ac.uk/medicine/FASTER/FPI/

Reviewer 2 Report

 - As described in the Statistical Analysis, an image with the histogram and boxplots could be presented, improving the reading and understanding of the manuscript results.

 - Results could include the load cell acquisition signals.

Author Response

POINT

REVIEWER COMMENT

RESPONSE

Reviewer #2

1

As described in the Statistical Analysis, an image with the histogram and boxplots could be presented, improving the reading and understanding of the manuscript results.

We inserted the figure as suggested (page 6):

Figure 3. Boxplots and histogram of the variables: (A) Foot Posture Index (FPI-6); (B) Normalized Foot Length; (C) Normalized Foot Strength; (D) Foot Arch Height – WB; (E) Midfoot Width – WB; (F) Foot Arch Height – NWB; (G) Midfoot Width – NWB; (H) DiffFAH; (I) DiffMFW; and (J) General

Foot Mobility. The violin plots show the data distribution, and the boxplots present the data locality, spread and skewness of dominant (black) and non-dominant (grey) feet.

2

Results could include the load cell acquisition signals.

Thank you for the suggestion. Unfortunately, the data registered from our load cell only computed peak forces, so we are unable to get full load cell plots. Still, the Normalized Foot Strength was calculated and used in previous studies to quantify the foot strength.

Reviewer 3 Report

It has been long speculated that foot intrinsic muscles and its mobility functions are highly correlated. The present study investigated this topic based on 24 recreational runners’ data. While the study was interesting, the main issue remains as demonstrating importance of the study. If little relationship was identified between hypothesised foot intrinsic muscles and its mobility functions, elaborate discussions on what else could be the factors. Please see further comments below.

Abstract: The importance of the study should be highlighted. Why were the foot intrinsic muscles and mobility important for recreational runners?

Line 21-22: ‘during the stance phase’ What about the swing phase?

Line 38-43: This part is misleading because it sounds as if the pronated foot is always bad. Pronation is a necessary function in gait but if done in the wrong timing or excessively, it could cause problems. Please do not create an impression that pronation should be avoided throughout.

Line 119: ‘normalized by the participant´s height (*100)’ What does this mean (i.e. *100)?

Discussion: Overall, fundamental improvements should be required. Explain why no association was observed. What controls foot mobility if not foot intrinsic strength? What implications can be provided for treatment options?

Author Response

POINT

REVIEWER COMMENT

RESPONSE

Reviewer #3

1

It has been long speculated that foot intrinsic muscles and its mobility functions are highly correlated. The present study investigated this topic based on 24 recreational runners' data. While the study was interesting, the main issue remains as demonstrating importance of the study. If little relationship was identified between hypothesised foot intrinsic muscles and its mobility functions, elaborate discussions on what else could be the factors. Please see further comments below.

Thank you for the critical considerations; the review helped improve our article's quality. We have edited several changes in the article. In particular, we inserted changes in the discussion section to present factors of the small correlation identified between hypothesized foot intrinsic muscles and its mobility functions

2

Abstract: The importance of the study should be highlighted. Why were the foot intrinsic muscles and mobility important for recreational runners?

We appreciate the suggestion. We have highlighted the study's importance in the abstract, which now reads:

Different measurements of foot morphological characteristics can effectively predict foot muscle strength. However, it is still uncertain if structural and postural alterations leading to foot pronation could be compensated with a more efficient function of the foot intrinsic muscles strength and how mobility and strength are associated. Also, the relationship between foot mobility and the strength of the intrinsic muscles that control the foot arch is still unclear. Therefore, this study aimed to investigate the morphological parameters between dominant and non-dominant feet and the relationship between the intrinsic foot muscle strength and foot mobility in recreational runners.

3

Line 21-22: 'during the stance phase' What about the swing phase?

We are aware that the swing phase is fundamental for running performance even though no forces are directly applied to the swinging/advancing foot by the ground. For example, the leg in the swing phase is essential to augment the vertical ground force passive impact peak and increase the horizontal displacement of the mass centre. Likely, the increases in early stance forces that are associated with the swing leg technique contribute to the asymmetrical vertical ground reaction force (Rottier & Allen, 2021). Therefore, we inserted comments regarding the swing phase (page 1). It now reads:

Running-related musculoskeletal pain affects 22% of runners [1], with the ankle-foot complex, in particular, representing 26% of all these injuries reported by runners [2]. As the first component of the kinetic chain to touch the ground during running, the human foot must be stiff enough during foot-strike, push-off, and mobile/compliant during the stance phase [3]. In addition, the swing leg is also fundamental to running performance. The swinging limbs can improve performance in jumping and running since it is conceivable that runners could use their swing leg to increase vertical ground force in the same way as an arm swing in vertical jumping [4].

Rottier, T. D., & Allen, S. J. (2021). The influence of swing leg technique on maximum running speed. Journal of Biomechanics, 126. (doi: 10.1016/j.jbiomech.2021.110640).

4

Line 38-43: This part is misleading because it sounds as if the pronated foot is always bad. Pronation is a necessary function in gait but if done at the wrong timing or excessively, it could cause problems. Please do not create an impression that pronation should be avoided throughout.

We have edited the sentence and addressed the importance of pronation. It now reads (page 2):

Furthermore, the medial longitudinal arch can allow for an energy-efficient running by acting as a spring (or a rubber ball) absorbing and storing elastic strain energy from the body during the first half of the stance phase of running and returning it during the second half of the stance [8]. Therefore, pronation is a necessary function in gait. However, if done excessively, the structural and morphological alterations in the longitudinal arch can lead to greater pronation of the foot and, therefore, dysfunctions. As an example, flatfoot can lead to poor motor skills and physical performance in children [9].

5

Line 119: 'normalized by the participant's height (*100)' What does this mean (i.e. *100)?

We apologize for this misunderstanding. The foot length normalization by the participant's height has been used in previous studies (Myer et al., 2013; Williams & McClay, 2000). After the normalization, the values are between 0-1 (or 0-100% if we multiply by 100). However, for further clarification, we have changed the sentence, which now reads (page 4):

The distance of the most posterior aspect of the calcaneus to the most distal point of the foot (first or the second toe, depending on which was longer) represented the foot length. This foot length was normalized by the participant’s height [33,34]. Therefore, the foot dorsum was demarcated as 50% of the foot length.

Myer, G. D., Sugimoto, D., Thomas, S., & Hewett, T. E. (2013). The influence of age on the effectiveness of neuromuscular training to reduce anterior cruciate ligament injury in female athletes: A meta-analysis. American Journal of Sports Medicine, 41(1), 203–215. (doi: 10.1177/0363546512460637).

Williams, D. S., & McClay, I. S. (2000). Measurements Used to Characterize the Foot and the Medial Longitudinal Arch: Reliability and Validity. Physical Therapy, 80(9).

6

Regarding the no correlation between intrinsic muscles and foot strength, different aspects could be related to this finding. The structure is not directly associated with the function, and mobility depends on tissue flexibility, joint movement (arthrokinematics) and motor control. For example, flexibility training using static stretching or proprioceptive neuromuscular facilitation stretching effectively enhances mobility.

Thank you for the suggestion. We have improved the discussion section.

Regarding the no correlation between intrinsic muscles and foot strength, different aspects could be related to this finding. The structure is not directly associated with the function, and mobility depends on tissue flexibility, joint movement (arthrokinematics) and motor control. Therefore, as a treatment option, flexibility training, such as static stretching or proprioceptive neuromuscular facilitation stretching, is suggested to effectively enhance mobility (Medeiros & Martini, 2018).

Medeiros, D. M., & Martini, T. F. (2018). Chronic effect of different types of stretching on ankle dorsiflexion range of motion: Systematic review and meta-analysis. Em Foot (Vol. 34, p. 28–35). Churchill Livingstone. (doi: 10.1016/j.foot.2017.09.006).

Reviewer 4 Report

From my point of view I see several inconsistencies: 1) The title does not respond to the study carried out, nor does the question stated at the end of the introduction. Both must be changed to correspond to the study that they have actually carried out. Since in the methodology they have only assessed the intrinsic Flexor Corpus muscle of the first toe and no other movement made by any of the 19 intrinsic muscles of the foot. 2) In the Introduction, it is necessary to talk about the intrinsic muscles of the foot, to name their functions in general and particular to each one or each muscle block depending on whether it is dorsal or plantar. 3) Figure 2 shows that they are measuring the mobility of the foot and are actually making the measurements of the morphology of the foot and anthropometric measurements. This is a bug and should be changed. It gives the impression that they did the study to assess the morphology of the foot and then it has been adapted to assess the mobility of the intrinsic muscles, but without carrying out said examination, only changing the objective and the title. 4) In the keywords, the word intrinsic musculature of the foot must appear. 5) I do not understand how the FPI gives results when in the objective of the study the posture of the foot is not considered as a question or variable, but the mobility of the foot and the intrinsic musculature, for which I do not find anything in the result that relates it

-Regarding the references in general, they are not updated, there are quite a few that are more than 10 years old.

-In the inclusion criteria there is a criterion that, when putting the exclusion criteria, shows that it has not been taken into account: Being free of musculoskeletal alterations in extremities. Taking into account this inclusion criterion, tendinitis and knee pathologies do not correspond to the exclusion criteria. This must be corrected.

-I detect an error in the bibliographic reference, reference number 2 and number 3, review both because both articles are presented with the same title.

-Reference number 18 has missing data such as the title, the same in reference number 21. In reference number 19 the year appears twice and not the number of pages. The references are wrong and should be corrected.

-In abbreviations an error also appears, in the ICC abbreviation as ICC Intra-rater reliability, when that is, Intraclass correlation coefficients. It must also be put on correctly.

Author Response

POINT

REVIEWER COMMENT

RESPONSE

Reviewer #4

1

The title does not respond to the study carried out, nor does the question stated at the end of the introduction. Both must be changed to correspond to the study that they have carried out. Since in the methodology they have only assessed the intrinsic Flexor Corpus muscle of the first toe and no other movement made by any of the 19 intrinsic muscles of the foot.

As recommended, we have made the manuscript's title and goal more specific to the measurements we have conducted.

New title:

Association between the Strength of Flexor Hallucis Brevis and Abductor Hallucis and Foot Mobility in Recreational Runners

Page 3, Lines 89-91: This cross-sectional study aimed to investigate whether an association exists between intrinsic foot muscle strength and foot mobility in recreational runners who are free from injuries.

2

In the Introduction, it is necessary to talk about the intrinsic muscles of the foot, to name their functions in general and particular to each one or each muscle block depending on whether it is dorsal or plantar.

We inserted a description of the intrinsic feet muscles.

It now reads (page 2):

The intrinsic muscles of the feet are active in dorsal and plantar functions. The intrinsic plantar muscles are commonly described based on their functional link with the longitudinal and transverse arches [14]. These muscles are composed of four layers of deep muscles in the plantar aponeurosis. The first two layers have muscle configurations that align with the foot's medial and lateral longitudinal arches [3]. Therefore, the intrinsic plantar muscles are functionally linked with the transverse and longitudinal medial arch [15]. Although the intrinsic muscles are more active in dynamic activities (e.g., walking) than standing [16], training these muscles enhances foot posture [17]. In addition, intrinsic foot muscles provide dynamic arch support during the propulsive phase of gait [18] and support the foot as a platform for standing and a lever for propelling the body during dynamic activities [19 ]. When considering running, the intrinsic muscles can stiffen the longitudinal arch during the push-off phase [20] to control foot posture [21]. Further, these muscles can guarantee an efficient transference of power generated by the ankle flexors soleus and gastrocnemius.

3

Figure 2 shows that they are measuring the mobility of the foot and are actually making the measurements of the morphology of the foot and anthropometric measurements. This is a bug and should be changed. It gives the impression that they did the study to assess the morphology of the foot and then it has been adapted to assess the mobility of the intrinsic muscles, but without carrying out said examination, only changing the objective and the title.

Thank you for this important suggestion. The anthropometric measurements described in Figure 2 were used to quantify the foot mobility magnitude, as suggested by McPoil et al. (2011). However, the caption has been edited to enhance clarification. It now reads:

Figure 2. Morphology and anthropometric measurements to quantify the foot mobility magnitude [27]: (a) midfoot width in weight-bearing; (b) midfoot width in non-weight-bearing; (c) foot arch height in weight-bearing ; (d) foot arch height in non-weight-bearing

4

In the keywords, the word intrinsic musculature of the foot must appear.

The keyword was included.

5

I do not understand how the FPI gives results when in the objective of the study the posture of the foot is not considered as a question or variable, but the mobility of the foot and the intrinsic musculature, for which I do not find anything in the result that relates it.

We agree with this critical concern. Thank you for the question. The Foot Posture Index (FPI) quantifies the degree to which a foot can be considered pronated, supinated or neutral. Therefore, we only utilized FPI to characterize the sample, as presented in Table 2.

6

Regarding the references in general, they are not updated, there are quite a few that are more than 10 years old.

Thank you for the comment. All references were reviewed, and more updated references have been inserted. Of the 39 citations, more than 60% were published in the past five years, and only seven are more than ten years old. These ten references are classical works and pertinent to our study, such as Ker, et al., (1982), published in Nature, where the authors presented the elastic properties of the arch of the human foot.

R. F., Bennett, M. B., Bibbyt, S. R., Kestert, R. C., & MeN Alexander, R. (1982). The spring in the arch of the human foot. Nature, 325, 7684–7689.

7

In the inclusion criteria, there is a criterion that, when putting the exclusion criteria, shows that it has not been taken into account: Being free of musculoskeletal alterations in extremities. Taking into account this inclusion criterion, tendinitis and knee pathologies do not correspond to the exclusion criteria. This must be corrected.

The inclusions/exclusion criteria have been rewritten. It now reads (page 3):

This cross-sectional and correlational study investigated 24 (18 men and 6 women) recreational runners selected via social media announcements. In addition, the principal investigator performed previous telephone screenings of potentially eligible participants. Study inclusion criteria involved 18-45 years of age, running at least 15 kilometres/week, and no history of ligament laxity, meniscal pathology, patellar tendonitis, knee pain from acute trauma, patellar dislocation and lower limb surgeries. Women should not be pregnant during the data acquisition. The Ethics Committee approved the study performed according to the Declaration of Helsinki. All participants provided written informed consent to participate.

8

I detect an error in the bibliographic reference, reference number 2 and number 3, review both because both articles are presented with the same title.

We apologize for this mistake. All references were reviewed and are now correct.

9

Reference number 18 has missing data such as the title, the same in reference number 21. In reference number 19 the year appears twice and not the number of pages. The references are wrong and should be corrected.

Thank you for catching these mistakes. We apologize for this reference oversight. All references have been revised.

10

In abbreviations an error also appears, in the ICC abbreviation as ICC Intra-rater reliability, when that is, Intraclass correlation coefficients. It must also be put on correctly.

We also apologize for this mistake. All the abbreviations have been reviewed and edited accordingly.

Round 2

Reviewer 1 Report

 The reviewer evaluates this paper deserves to be published in Biomechanics, which will contribute the increase in our information about the foot intrinsic muscles strength and the foot mobility of runners.

Reviewer 3 Report

The authors have addressed all my comments sufficiently. I have no further comments.

Reviewer 4 Report

Thank you for allowing me to verify the subsequent adequacy of this article after the modifications made. From my point of view, everything has been correctly corrected and is fit for publication.

Sincerely